# Context-Length Robustness in Question Answering Models: A Comparative Empirical Study

**Trishita Dhara**
Upper Hand
trishitadhara123@gmail.com

**Siddhesh Sheth**
Ace Rent a Car
shethsiddhesh268@gmail.com

## Abstract

Large language models are increasingly deployed in settings where relevant information is embedded within long and noisy contexts. Despite this, robustness to growing context length remains poorly understood across different question answering tasks. In this work, we present a controlled empirical study of context-length robustness in large language models using two widely used benchmarks: SQuAD and HotpotQA.

We evaluate model accuracy as a function of total context length by systematically increasing the amount of irrelevant context while preserving the answer-bearing signal. This allows us to isolate the effect of context length from changes in task difficulty. Our results show a consistent degradation in performance as context length increases, with substantially larger drops observed on multi-hop reasoning tasks compared to single-span extraction tasks. In particular, HotpotQA exhibits nearly twice the accuracy degradation of SQuAD under equivalent context expansions.

These findings highlight task-dependent differences in robustness and suggest that multi-hop reasoning is especially vulnerable to context dilution. We argue that context-length robustness should be evaluated explicitly when assessing model reliability, especially for applications involving long documents or retrieval-augmented generation.

## 1 Introduction

Large language models have recently achieved strong performance on a wide range of question answering tasks, including both extractive and multi-hop reasoning benchmarks Devlin et al. (2019); Brown et al. (2020). In many practical applications, however, these models operate over long and heterogeneous contexts that contain a mixture of relevant and irrelevant information. Examples include retrieval-augmented generation, document analysis, and multi-document question answering systems. In such settings, models must reliably identify and reason over a small subset of context while ignoring distractors.

Despite growing interest in long-context modeling Beltagy et al. (2020); Dao et al. (2022), most evaluations of question answering systems assume a fixed or moderately sized input context. This assumption obscures an important aspect of model reliability: how performance changes as the total amount of available context increases while the underlying question and answer remain unchanged. As context length grows, additional irrelevant content may dilute the signal required for correct reasoning, potentially leading to systematic performance degradation.

In this work, we study *context-length robustness* in large language models through a controlled empirical evaluation. Rather than altering question difficulty or answer ambiguity, we construct evaluation settings in which the answer-bearing content is preserved and progressively embedded within longer contexts by injecting additional irrelevant text. This design allows us to isolate the effect of context length itself from other confounding factors, such as changes in prompt formulation or task structure.

We conduct experiments on two widely used question answering benchmarks with distinct characteristics: SQuAD Rajpurkar et al. (2016), which primarily involves single-span answer extraction from a given passage, and HotpotQA Yang et al. (2018), which requires multi-hop reasoning across multiple pieces of evidence. By applying equivalent context-length expansions to both datasets, we analyze how robustness to increasing context length varies across task types.

Our results show that model accuracy consistently degrades as context length increases, even when the relevant information required to answer the question is retained. Moreover, the magnitude of degradation differs substantially across tasks: multi-hop reasoning exhibits significantly higher sensitivity to context expansion than single-span extraction. These findings indicate that robustness to long and noisy contexts is task-dependent and suggest that context-length robustness should be evaluated explicitly when assessing the reliability of question answering models.

## 2 RELATED WORK

Recent research has increasingly focused on the effects of context length on language model performance. Theoretical work by Shi et al. Shi et al. (2025) proposes a framework explaining how intrinsic dimension and training dataset size influence optimal context length and the scaling behavior of language models. Complementary surveys, such as Pawar et al. Pawar et al. (2024), provide comprehensive overviews of techniques proposed to support extended input sequences in transformers.

Contemporaneous empirical investigations have demonstrated that extended context windows do not always translate into improved performance. For example, Du et al. Du et al. (2025) reveal that even with perfect retrieval of relevant evidence, LLM performance can decline as context length increases—a finding closely aligned with the robustness trends observed in our study. These results suggest that context length alone, independent of retrieval quality, can hinder reasoning performance.

Beyond formal research publications, practitioners and industry observers have emphasized the practical importance and limitations of context length in real-world models. Articles such as those by Moesker Moesker (2024) and exploratory work on context rotation strategies Team (2025) highlight how context window design and token ordering can influence model behavior, underscoring the need for systematic robustness evaluations. Together, this body of work motivates our focus on quantifying the sensitivity of model accuracy to increasing context lengths across QA datasets and model families.

**Empirical analyses of long-context limitations.** Multiple studies have shown that LLM performance degrades as context length increases, even when the answer-relevant information is present. Previous researches demonstrate that language models often fail to exploit long-range dependencies in practice, despite architectural capacity to process long sequences. Li et al. Li et al. (2024) further observe that models struggle with long in-context learning, exhibiting position-dependent performance degradation commonly referred to as the "lost-in-the-middle" effect. These findings suggest that long context availability does not necessarily translate into robust reasoning over extended inputs.

**Benchmarks for long-context understanding.** Several benchmarks have been proposed to systematically evaluate long-context capabilities. BABILong Kuratov et al. (2024) tests reasoning performance across contexts ranging from thousands to millions of tokens and shows that most models fail beyond relatively modest lengths. XL$^2$Bench Ni et al. (2024) evaluates extremely long documents across diverse tasks such as memory retrieval, detailed understanding, and open-ended generation, revealing substantial gaps between model performance and human-level comprehension. While these benchmarks provide valuable stress tests, they often combine multiple sources of difficulty, including task complexity, retrieval, and context length, making it challenging to isolate the specific effect of context expansion.

**Context window extension methods.** Another line of work focuses on extending the maximum context window of LLMs. Position interpolation Chen et al. (2023) enables context length extrapolation with limited fine-tuning and has been shown to reduce perplexity degradation at longer lengths.

Subsequent approaches, including RoPE-based extensions and related interpolation techniques, further improve stability when processing longer sequences. However, these methods primarily address architectural or positional encoding constraints and do not directly evaluate whether models remain robust to irrelevant or distracting context once the window is extended.

**Prompt compression and retrieval-based approaches.**    To mitigate the computational and performance challenges of long contexts, several studies propose prompt compression or retrieval-based strategies. LongLLMLingua Jiang et al. (2024) introduces question-aware prompt compression to reduce irrelevant information and alleviate position bias, achieving significant gains on long-context benchmarks. Similarly, retrieval-augmented generation has been shown to recover performance using shorter effective contexts Xu et al. (2024). However, these approaches alter the input distribution by selectively removing or reordering information, making it difficult to assess intrinsic robustness to long context itself.

**Robustness and reasoning under interference.**    Recent work highlights that long-context failures are often tied to reasoning complexity rather than sheer length. Kuratov et al. Kuratov et al. (2024) show that performance drops sharply for tasks requiring multi-hop reasoning, even when single-fact retrieval remains reliable. Related findings indicate that LLMs encode long-range context superficially, with limited sensitivity to semantic structure beyond a few thousand tokens. These observations motivate evaluations that control for evidence availability while varying only the amount of irrelevant context.

**Positioning of this work.**    In contrast to prior benchmarks and architectural studies, our work focuses on a controlled robustness evaluation in which the answer-bearing signal is explicitly preserved across all context lengths. By systematically injecting irrelevant context while holding task difficulty and evidence constant, we isolate the effect of context length on model performance. Our evaluation across both single-hop (SQuAD) and multi-hop (HotpotQA) question answering reveals task-dependent robustness patterns that complement existing long-context benchmarks. Rather than proposing new architectures or compression techniques, we aim to provide a clearer empirical characterization of when and how context length interferes with reasoning in current LLMs.

## 3 METHODOLOGY

This section describes the datasets, context construction procedure, evaluation protocol, and parameter settings used to study context-length robustness. All design choices are made to isolate the effect of context length while maintaining reproducibility and comparability across models and tasks.

### 3.1 DATASETS

We evaluate context-length robustness on two widely used question answering benchmarks with distinct reasoning characteristics.

**SQuAD.**    The Stanford Question Answering Dataset (SQuAD) Rajpurkar et al. (2016) is an extractive question answering benchmark in which answers correspond to contiguous spans within a single passage. SQuAD primarily tests localized evidence extraction and reading comprehension. Its relatively focused structure makes it well suited for evaluating robustness in settings where the answer can be derived from a small region of text.

**HotpotQA.**    HotpotQA Yang et al. (2018) is a multi-hop question answering benchmark that requires combining information from multiple supporting documents. We use the distractor setting of HotpotQA, which includes both relevant and irrelevant passages. This benchmark tests a model's ability to integrate evidence across documents and is therefore more sensitive to interference from irrelevant context.

For both datasets, we use the validation splits and evaluate a fixed subset of $n = 200$ instances. Using a shared subset across all context lengths and models ensures that observed performance differences are attributable to context length rather than sample variation. The sample size is chosen

to balance statistical stability with computational cost; with $n = 200$, accuracy differences are quantized in increments of approximately $1.7\%$, which we account for when interpreting results.

## 3.2 CONTEXT CONSTRUCTION

Our evaluation focuses on isolating the effect of total context length while preserving the information necessary to answer each question. To achieve this, we construct contexts consisting of two components: a *signal* segment containing the answer-bearing information, and a set of *distractor* segments containing irrelevant text.

For each question, we first extract a signal window of up to 256 tokens from the original context that contains the gold answer. This window size is chosen to ensure that sufficient local evidence is retained for both single-span extraction and multi-hop reasoning, while minimizing the inclusion of unnecessary surrounding text. If the answer does not appear contiguously within a 256-token span, the earliest available span is used.

To increase total context length, we prepend distractor segments sampled from other instances within the same dataset. Distractors are constructed by chunking unrelated contexts into fixed-length segments of 128 tokens. To prevent answer leakage, any distractor segment containing the normalized gold answer string is excluded. Distractor pools are built globally per dataset and reused across models to ensure consistent perturbations.

We evaluate four target context lengths: 256, 512, 1024, and 2048 tokens. These values are selected to cover short, moderate, and long contexts commonly encountered in practical applications, while remaining within the effective operating range of the evaluated models. For each target length, the combined context is truncated or padded to match the specified token budget exactly. The answer-bearing signal is always preserved across all context lengths. Figure 1 illustrates the controlled context construction procedure used throughout our evaluation.

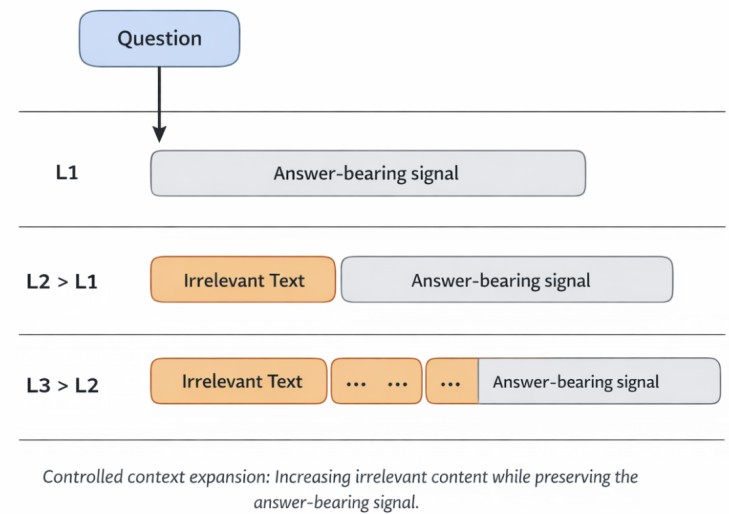

*Controlled context expansion: Increasing irrelevant content while preserving the answer-bearing signal.*

Figure 1: Controlled context construction used to evaluate context-length robustness. For each question, an answer-bearing signal segment is preserved across all settings, while increasing amounts of irrelevant context are added to reach target context lengths. This design isolates the effect of total context length while keeping the available evidence fixed.

## 3.3 MATHEMATICAL FORMULATION

We formalize context-length robustness as the sensitivity of model performance to changes in total input length under fixed task conditions. Let $M$ denote a question answering model, and let $\mathcal{D} = \{(q_i, a_i)\}_{i=1}^{n}$ denote a dataset of questions and reference answers. For each example, we construct a

context $c_i(L)$ of total token length $L$, such that the answer-bearing signal is preserved for all values of $L$.

We define model accuracy at context length $L$ as

$$\text{Acc}_M(L) = \frac{1}{n} \sum_{i=1}^{n} \mathbf{1}\big(M(q_i, c_i(L)) = a_i\big), \tag{1}$$

where $\mathbf{1}(\cdot)$ denotes the indicator function.

Context-length robustness is characterized by the behavior of $\text{Acc}_M(L)$ as a function of $L$. To quantify degradation, we define the absolute robustness drop between two context lengths $L_1 < L_2$ as

$$\Delta_M(L_1, L_2) = \text{Acc}_M(L_1) - \text{Acc}_M(L_2). \tag{2}$$

This formulation allows us to compare robustness across models and tasks by analyzing differences in the accuracy–length curves $\text{Acc}_M(L)$. In particular, larger values of $\Delta_M$ indicate greater sensitivity to increasing context length under otherwise fixed conditions.

### 3.3.1 A Signal-to-Noise Perspective on Context Degradation

Let each constructed context be decomposed as $c_i(L) = s_i \oplus n_i(L)$, where $s_i$ is the fixed answer-bearing signal window and $n_i(L)$ is irrelevant distractor content chosen such that the total context length is $|c_i(L)| = L$. Since transformer attention is normalized across the available context, a simple null model assumes that, absent strong priors, the expected attention mass assigned to the signal is proportional to its share of tokens:

$$\mathbb{E}[\alpha_s(L)] \approx \frac{|s_i|}{L}. \tag{3}$$

As $L$ increases while $|s_i|$ remains fixed, the expected signal mass decreases, yielding an effective signal-to-noise degradation. For multi-hop tasks requiring aggregation across multiple evidence fragments, this effect compounds across reasoning steps: if a solution requires $k$ successful selections or integrations of relevant evidence, a crude approximation gives an overall success probability scaling like

$$p_k(L) \propto \Big(\mathbb{E}[\alpha_s(L)]\Big)^k, \tag{4}$$

which decays faster in $L$ for larger $k$. Although simplified, this framing provides a mechanistic explanation consistent with our empirical observation that HotpotQA degrades more sharply than SQuAD under increasing irrelevant context.

## 3.4 Models

We evaluate two instruction-tuned large language models with different capacity profiles: `gpt-4.1` and `gpt-4.1-mini`. This pairing allows us to examine whether context-length robustness trends are consistent across model scales. All models are queried using identical prompts and decoding settings to ensure fair comparison.

## 3.5 Evaluation Protocol

For each model, dataset, and target context length, we prompt the model to answer the question using the constructed context. We use deterministic decoding with temperature set to zero to eliminate variability due to sampling and to ensure reproducibility across runs.

Model predictions are evaluated using exact match (EM), which measures whether the predicted answer exactly matches the reference answer after standard normalization. Accuracy is computed independently at each context length and then averaged across all evaluated instances. To quantify robustness, we additionally report the absolute accuracy drop between the shortest and longest context settings.

To avoid rate-limit artifacts and partial evaluations, all model queries are executed with enforced inter-request delays and exponential backoff. Failed queries are excluded from caching and retried to ensure uniform coverage across all experimental conditions. In the final results, no failed queries are observed.

## 3.6 IMPLEMENTATION DETAILS

Token lengths are measured using the tokenizer associated with the evaluated models. Distractor sampling is performed deterministically using fixed random seeds to ensure that the same distractor sets are used across reruns and across models. All results are cached at the level of individual model–dataset–context-length combinations to prevent accidental recomputation and to maintain consistency when experiments are resumed.

All hyperparameters used in the evaluation are summarized in Table 1.

| Parameter | Value |
|---|---|
| Signal window size | 256 tokens |
| Distractor chunk size | 128 tokens |
| Target context lengths | 256, 512, 1024, 2048 |
| Number of examples | 200 per dataset |
| Decoding temperature | 0.0 |
| Evaluation metric | Exact Match (EM) |

Table 1: Summary of evaluation parameters.

We evaluate 200 validation instances per dataset and report 95% bootstrap confidence intervals. At this scale, the uncertainty is sufficiently narrow to clearly distinguish the substantial degradation observed on HotpotQA from the comparatively stable behavior on SQuAD.

## 4 RESULTS

We evaluate context-length robustness by measuring exact match accuracy as a function of total context length across datasets and model variants. Our analysis focuses on two questions: (i) how performance degrades as irrelevant context increases, and (ii) whether robustness patterns differ across task types and model scales.

## 4.1 OVERALL ROBUSTNESS TRENDS

Figure 2 visualizes the accuracy–length relationship both in terms of mean performance and associated uncertainty, confirming that the observed degradation trends are robust to sampling variability.

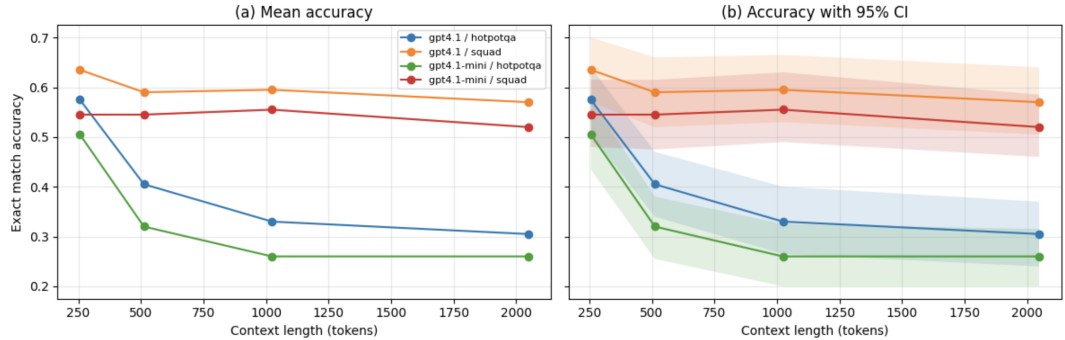

Figure 2: Context-length robustness across tasks and models. (a) Mean exact match accuracy as a function of total context length. (b) Accuracy with 95% bootstrap confidence intervals computed over evaluation instances. Performance degradation on HotpotQA remains consistent under uncertainty, while SQuAD exhibits comparatively stable behavior across context lengths.

Across all settings, increasing context length leads to a non-increasing trend in accuracy. However, the magnitude of degradation differs substantially between datasets. HotpotQA exhibits pronounced sensitivity to context expansion, while SQuAD remains comparatively stable even at longer context lengths.

## 4.2 TASK-DEPENDENT DEGRADATION

Table 2 reports exact match accuracy at each evaluated context length. For HotpotQA, accuracy decreases sharply as context length increases from 256 to 1024 tokens, after which performance largely saturates. In contrast, SQuAD shows only minor fluctuations across the same range.

| Model | Dataset | 256 | 512 | 1024 | 2048 |
|---|---|---|---|---|---|
| gpt-4.1 | HotpotQA | 0.575 | 0.405 | 0.330 | 0.305 |
| gpt-4.1-mini | HotpotQA | 0.505 | 0.320 | 0.260 | 0.260 |
| gpt-4.1 | SQuAD | 0.635 | 0.590 | 0.595 | 0.570 |
| gpt-4.1-mini | SQuAD | 0.545 | 0.545 | 0.555 | 0.520 |

Table 2: Exact match accuracy at different context lengths for each dataset and model. Results are averaged over 200 validation instances per dataset.

These results indicate that multi-hop reasoning is substantially more vulnerable to interference from irrelevant context than single-span extraction. While SQuAD accuracy varies within a narrow range across context lengths, HotpotQA accuracy drops by more than 20 percentage points in some settings.

## 4.3 EFFECT OF MODEL SCALE

To assess whether robustness trends depend on model capacity, we compare gpt-4.1 and gpt-4.1-mini under identical evaluation conditions. For both datasets, the larger model exhibits consistently higher accuracy across all context lengths. However, the relative degradation pattern remains similar across models.

Table 3 reports the absolute accuracy drop between the shortest (256 tokens) and longest (2048 tokens) context settings. On HotpotQA, both models experience substantial degradation, with the smaller model exhibiting a slightly larger drop. On SQuAD, degradation is minimal for both models.

| Model | Dataset | Accuracy Drop (256 $\rightarrow$ 2048) |
|---|---|---|
| gpt-4.1 | HotpotQA | 0.270 |
| gpt-4.1-mini | HotpotQA | 0.245 |
| gpt-4.1 | SQuAD | 0.065 |
| gpt-4.1-mini | SQuAD | 0.025 |

Table 3: Absolute exact match accuracy drop from the shortest (256 tokens) to the longest (2048 tokens) evaluated context length. Results are averaged over 200 validation instances per dataset.

These findings suggest that increasing model capacity improves overall performance but does not eliminate sensitivity to long and noisy contexts, particularly for tasks requiring multi-hop reasoning.

## 5 DISCUSSION

Our results provide a quantitative characterization of context-length robustness in large language models under controlled perturbations. By analyzing the accuracy–length function defined in Section 3.3, we observe consistent performance degradation as irrelevant context increases, even when answer-bearing evidence is preserved. This confirms that extended context availability alone does not guarantee robust utilization of relevant information.

A key finding is the pronounced task-dependent nature of robustness. On HotpotQA, which requires multi-hop reasoning across multiple pieces of evidence, accuracy drops sharply as context length increases. In contrast, SQuAD exhibits relatively stable performance across the same context expansions. This suggests that reasoning complexity, rather than context length alone, plays a central role in determining robustness. When answering requires integrating multiple facts, interference from irrelevant context appears to disproportionately affect model performance.

We also observe that increasing model capacity improves absolute accuracy but does not fundamentally alter robustness trends. While gpt-4.1 consistently outperforms gpt-4.1-mini, both

models exhibit similar degradation patterns as context length increases. This indicates that context-length sensitivity is not solely a consequence of limited model capacity, but may instead reflect structural limitations in how current models attend to and integrate information under noisy conditions.

From a mathematical perspective, these findings highlight that the accuracy function $\text{Acc}_M(L)$ is not invariant to context expansion and that its rate of decay varies substantially across tasks. This reinforces the importance of evaluating robustness as a function rather than relying on single-point performance measurements at fixed context lengths. Such functional analysis provides a more informative view of model reliability in realistic settings where available context may vary.

## 6 LIMITATIONS

Our study has several limitations. First, the evaluation is restricted to two question answering benchmarks and two model variants. While these choices are sufficient to demonstrate task-dependent robustness patterns, broader conclusions would require evaluation across additional tasks, domains, and architectures. Second, we consider context lengths up to 2048 tokens; although this range covers many practical applications, future work could extend the analysis to substantially longer contexts.

Additionally, our robustness metric is based on exact match accuracy, which may not capture partial correctness or reasoning quality. Finally, while our controlled context construction isolates the effect of irrelevant context length, it does not model all real-world sources of noise, such as contradictory or adversarial information. These limitations suggest directions for extending the proposed evaluation framework rather than detracting from the core findings. While our evaluation uses a fixed subset of 200 instances per dataset, we report confidence intervals to mitigate sampling variability. Larger-scale evaluations are a natural direction for future work.

## 7 CONCLUSION

In this work, we presented a controlled empirical study of context-length robustness in large language models. By formalizing accuracy as a function of total context length and systematically injecting irrelevant content while preserving answer-bearing evidence, we isolated the effect of context expansion on question answering performance.

Our experiments demonstrate that robustness to increasing context length is strongly task-dependent. Multi-hop reasoning tasks exhibit substantial performance degradation as context grows, whereas single-span extraction remains comparatively stable. Increasing model capacity improves overall accuracy but does not eliminate sensitivity to long and noisy contexts.

These findings underscore the importance of evaluating robustness explicitly when assessing model reliability in long-context settings. We hope that the proposed formulation and evaluation protocol will serve as a foundation for future work on understanding and improving the robustness of language models under realistic context conditions.

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
