# OpenReview forum: "Context-Length Robustness in Question Answering Models: A Comparative Empirical Study"
_mathai.club/MathAI/2026/Conference — 2026 Oral_

### Official Review · Reviewer_MWmy · 2026-03-12
**Reject: The incomplete study with missed opportunities to go beyond experiments and heuristics**

**Rating:** 3
**Confidence:** 5

**Review:**

$Originality/Novelty$

The authors note: "Our results show a consistent degradation in performance as context length increases". That is known from a few theoretical results via complexity analysis and statistics. A similar argument applies to the statement: "...multi-hop reasoning is especially vulnerable to context dilution".

The results do not seem novel or original (the benchmarks appear old). Modern architectures are trying to compensate for the degradation phenomenon with long-running agentic architectures. That conjecture is out of the scope of the paper. Also, in the context of the paper, one has to distinguish between a rigorous context (e.g., when the task is to prove a statement) and a well-formulated goal with concise scaffolding for a multi-step solution. Thus, an agentic experimentation is more appropriate for the contemporary study the authors engage with.

The distractor technique is also not novel.

$Soundness;Significance$
Due to a lack of novelty, significance is diminished.
There is no reason to question the experimental results, despite the crude estimation in a rare theoretical considerations of Section 3.3.1:
 $\mathbb{E}[α_s(L)]≈ |s_i|/L$ where $s_i$ are signals, L is the length of the context, and $\alpha_s$ is (the expected) attention mass assigned to the signal.

The standard analysis, based on, say, an estimate of generalization via length and Rademacher dimension, would be appropriate for this; it seems only technical to conduct—missed opportunity.

$Presentation$
The exposition is clear and fairly concise. It is a strength.

$Questions$

1. Section 3.6. Note that the temperature in Table 1 is zero, so there is no randomization. Isn't that further masking the actual performance, from a practical standpoint?

2. Section 4.2. The models are outdated. Without Claude, this is incomplete & missing the acceptable baseline. Also, since 0-1 accuracy is adopted, why not use cross-entropy rather than exact match?

3. Section 4.3. An agentic CoT context is needed.

$Limitations$

The authors recognize the theoretical limitations. In general, that is a strength.
However, they noted that robustness metrics are based on exact-match accuracy, which may not capture partial correctness or reasoning quality (as we pointed out earlier, cross-entropy captures this).

$Weaknesses$:
1. Too limited a theoretical view.

2. Even a rudimentary, informal analysis is incomplete, which is not conducive to the premise of this conference.

3. Missed opportunity for a relatively easy degradation estimate under the conditions adopted - that leads to no theoretical novelty whatsoever.

4. Absence of the latest benchmarks and models.

$Strengths$

1. Clear and succinct exposition.

2. Good realization of limitations.

Unfortunately, the weaknesses overcome the strengths.

Hence, the conclusion.

---

### Official Review · Reviewer_dBJi · 2026-03-13
**Accept with major revision**

**Rating:** 6
**Confidence:** 4

**Review:**

**Summary**

This paper investigates the robustness of large language models (LLMs) to increasing context length in question answering tasks. The authors propose a controlled experimental approach in which answer-bearing information is preserved unchanged while the volume of irrelevant context is systematically increased. This design isolates the effect of context length from changes in task difficulty. Experiments are conducted on two benchmarks with distinct characteristics: SQuAD (single-span answer extraction) and HotpotQA (multi-hop reasoning). The main finding is that model performance monotonically degrades as context length increases, with degradation being substantially more pronounced for multi-hop reasoning tasks: HotpotQA demonstrates nearly twice the accuracy drop compared to SQuAD under equivalent context expansion. The paper emphasizes the need for explicit evaluation of context-length robustness when validating LLMs for applications involving long documents.

**Strengths**

1. Quality of experimental design. The work demonstrates a high methodological standard: the authors clearly formalize the notion of context-length robustness through the function Acc_M(L), define the metric of absolute accuracy drop Δ_M, and introduce a signal-to-noise perspective that provides a plausible mechanistic explanation for the observed effects. The controlled context construction procedure with preserved answer-bearing signal allows for valid isolation of the factor under investigation.
Statistical rigor. The authors correctly employ bootstrap confidence intervals (95% CI) to quantify uncertainty, which is particularly important given the limited sample size (n=200). The explicit mention of accuracy quantization (~1.7%) demonstrates the authors' awareness of the limitations of their experimental design.

2. Relevance to Trusted AI. The research directly relates to model robustness—a key area within MathAI. The discovered dependency between reasoning complexity and context sensitivity has practical implications for developing reliable RAG (Retrieval-Augmented Generation) systems and long-document analysis pipelines.

3. Clarity of presentation. The paper is written in an academic style with a logical and transparent structure. The Related Work section provides a comprehensive overview of adjacent research, including theoretical work on context scaling, long-context benchmarks, and context window extension methods. The positioning of the work relative to existing approaches is formulated appropriately.
Reproducibility considerations. Table 1 contains all key experimental hyperparameters (signal window size, distractor chunk size, target context lengths, decoding temperature). The authors explicitly indicate the use of deterministic distractor sampling with fixed random seeds.

**Weaknesses and Limitations**

1. Limited experimental scope. The study covers only two models from the same family (gpt-4.1 and gpt-4.1-mini) and two datasets. The absence of comparison with alternative architectures (e.g., Claude, Llama, Mistral) substantially limits the generality of the conclusions. The maximum context length of 2048 tokens does not reflect current practices: many models support 32K–128K tokens, and benchmarks such as BABILong test contexts up to 1M tokens.

2. Exclusive use of closed proprietary models. The authors' decision to use only OpenAI models (gpt-4.1 and gpt-4.1-mini) raises concerns regarding scientific reproducibility and transparency. Closed models do not allow researchers to independently verify results, analyze internal model representations (attention weights, hidden states), or reproduce experiments when the API changes. Experiments on open model families (Qwen, Gemma, Llama, Mistral) would be substantially more informative: they enable not only reproducibility verification but also deeper analysis of degradation mechanisms, including model introspection. Furthermore, open models allow version control and exclude the influence of hidden updates, which is critical for scientific validity.

3. Absence of multilingual analysis. Modern LLMs are predominantly multilingual, yet their training corpora are dominated by one or two languages (English, or English and Chinese). The authors did not investigate robustness for languages poorly represented in training corpora, although such experiments could demonstrate substantially different LLM behavior patterns on long contexts. In particular, for Russian—a language that can be classified as having relatively weak representation in the training data of modern multilingual LLMs—there exist well-developed benchmarks such as Libra, which analyze model behavior on very long input contexts. Notably, Libra includes BABILong, which the authors mention, yet the authors did not take advantage of the opportunity to compare their results from a multilingual perspective. Ignoring the language factor substantially reduces the generality of conclusions, since attention mechanisms and positional encoding may behave differently for languages with different morphology, word order, and tokenization.

4. Small sample size. Although n=200 provides statistical stability for detecting substantial effects (as shown on HotpotQA), this size may be insufficient for revealing more subtle patterns. For example, differences in degradation between models on SQuAD may be statistically insignificant at this scale. Additionally, accuracy is quantized in increments of ~1.7%, which coarsens the measurements.
Absence of public code and data. Despite detailed methodology description, the paper contains no links to a code repository or constructed datasets. This contradicts NeurIPS reproducibility standards and complicates result verification by independent researchers. The reproducibility checklist is not explicitly provided. This limitation is particularly critical in combination with the use of closed models: other researchers have no opportunity to either verify the authors' results or extend the experiments to other models or languages.
Exact Match metric. Using only EM does not allow assessment of partial answer quality or reasoning correctness. For HotpotQA, where answers may require information synthesis, F1-score or semantic similarity metrics would be more informative. Furthermore, EM does not distinguish between "almost correct" and "completely wrong" answers.

5. Observational nature without interpretation. Although the signal-to-noise model (equations 3–4) is proposed as an explanation, it remains hypothetical. The authors do not conduct attention analysis to verify this mechanism. Additionally, there is no discussion of whether positioning answer-bearing information in different parts of the context (beginning, middle, end) may affect results—the "lost-in-the-middle" effect is well known in the literature.

**Questions to Authors**

Q1. Choice of maximum context length.

Why was the range up to 2048 tokens chosen when modern LLMs support significantly longer contexts? Are there plans to extend the experiment to 8K–32K tokens, where degradation effects may manifest differently? Answering this question could raise the Significance score if the authors demonstrate that patterns persist at longer contexts.

Q2. Signal position in context.

Did the authors conduct experiments with different positioning of answer-bearing information (beginning, middle, end of context)? Existing literature indicates a "U-shaped" performance dependency on position. If not, is such analysis planned? This would substantially affect the Quality score.

Q3. Verification of signal-to-noise hypothesis.

Equations (3–4) describe a plausible mechanism but are not experimentally verified. Did the authors consider analyzing attention weights to confirm that attention to signal tokens indeed decays proportionally to their share in the context? This would strengthen the theoretical contribution of the paper.

Q4. Extension of model set.

Is inclusion of models with different architectures planned, especially open ones (Qwen, Gemma, Llama, Mistral), to verify the generality of conclusions? Using open models would enable introspection and ensure independent reductibility of results. It would be interesting to see whether the "multi-hop is more sensitive" pattern persists for models with different attention mechanisms or positional encoding.

Q6. Multilingual robustness.

Did the authors conduct experiments in languages other than English? In particular, were they interested in robustness for languages with weak representation in training corpora (e.g., Russian)? Are there plans for integration with multilingual benchmarks like Libra, which includes BABILong and is specifically designed for analyzing model behavior on long contexts across different languages?

---

### Decision · Program_Chairs · 2026-03-14

**Decision:**

Accept (Oral)

**Comment:**

Dear Author(s),

On behalf of the Program Committee of the International Conference on Mathematics of Artificial Intelligence (MathAI 2026), we are pleased to inform you that your paper has been accepted for an oral presentation at MathAI 2026.

Your paper was evaluated through a rigorous two-stage review process involving both automated screening and expert review by members of the Program Committee. The reviewers recognized the quality and contribution of your work.

Presentation details:

- Format: Oral presentation (15–20 minutes + 5 minutes Q&A)
- Mode: You may present either in person (offline) at the conference venue in Sirius, Russia, or remotely via Zoom. Please indicate your preferred mode when confirming your participation.
- Conference dates: Marh 30 - April 3, 2026
- Website: https://mathai.club

Next steps:

1. Please confirm your participation and presentation mode by replying to this email mathai.club@yandex.ru no later than March 15, 2026 18:00 Moscow time.
2. If you plan to attend in person, the organizing committee will provide accommodation details separately.
3. Please prepare your final camera-ready manuscript according to the formatting guidelines available at https://mathai.club and upload it to OpenReview by March 15, 2026 18:00 Moscow time.

Should you have any questions regarding the program, logistics, or your presentation slot, please do not hesitate to contact us.

We look forward to your contribution to MathAI 2026.

With kind regards,

MathAI 2026 Program Committee
International Conference on Mathematics of Artificial Intelligence
https://mathai.club
OpenReview: https://openreview.net/group?id=mathai.club/MathAI/2026/Conference
Telegram: https://t.me/MathAI_club
Email: mathai.club@yandex.ru